# WHEN DO CURRICULA WORK?

**Xiaoxia Wu**[*]
UChicago and TTIC
xwu@ttic.edu

**Ethan Dyer**
Blueshift, Alphabet
edyer@google.com

**Behnam Neyshabur**
Blueshift, Alphabet
neyshabur@google.com

## ABSTRACT

Inspired by human learning, researchers have proposed ordering examples during training based on their difficulty. Both curriculum learning, exposing a network to easier examples early in training, and anti-curriculum learning, showing the most difficult examples first, have been suggested as improvements to the standard i.i.d. training. In this work, we set out to investigate the relative benefits of ordered learning. We first investigate the *implicit curricula* resulting from architectural and optimization bias and find that samples are learned in a highly consistent order. Next, to quantify the benefit of *explicit curricula*, we conduct extensive experiments over thousands of orderings spanning three kinds of learning: curriculum, anti-curriculum, and random-curriculum – in which the size of the training dataset is dynamically increased over time, but the examples are randomly ordered. We find that for standard benchmark datasets, curricula have only marginal benefits, and that randomly ordered samples perform as well or better than curricula and anti-curricula, suggesting that any benefit is entirely due to the dynamic training set size. Inspired by common use cases of curriculum learning in practice, we investigate the role of limited training time budget and noisy data in the success of curriculum learning. Our experiments demonstrate that curriculum, but not anti-curriculum can indeed improve the performance either with limited training time budget or in existence of noisy data.[1]

## 1 INTRODUCTION

Inspired by the importance of properly ordering information when teaching humans (Avrahami et al., 1997), curriculum learning (CL) proposes training models by presenting easier examples earlier during training (Elman, 1993; Sanger, 1994; Bengio et al., 2009). Previous empirical studies have shown instances where curriculum learning can improve convergence speed and/or generalization in domains such as natural language processing (Cirik et al., 2016; Platanios et al., 2019), computer vision (Pentina et al., 2015; Sarafianos et al., 2017; Guo et al., 2018; Wang et al., 2019), and neural evolutionary computing (Zaremba & Sutskever, 2014). In contrast to curriculum learning, anti-curriculum learning selects the most difficult examples first and gradually exposes the model to easier ones. Though counter-intuitive, empirical experiments have suggested that anti-curriculum learning can be as good as or better than curriculum learning in certain scenarios (Kocmi & Bojar, 2017; Zhang et al., 2018; 2019b). This is in tension with experiments in other contexts, however, which demonstrate that anti-curricula under perform standard or curriculum training (Bengio et al., 2009; Hacohen & Weinshall, 2019).

As explained above, empirical observations on curricula appear to be in conflict. Moreover, despite a rich literature (see Section A), no ordered learning method is known to improve consistently across contexts, and curricula have not been widely adopted in machine learning. This suggest ruling out curricula as a beneficial practice for learning. In certain contexts, however, for large-scale text models such as GPT-3 (Brown et al., 2020) and T5 (Raffel et al., 2019), non-uniform mixing strategies are standard practice. These contradicting observations contribute to a confusing picture on the usefulness of curricula.

This work is an attempt to improve our understanding of curricula systematically. We start by asking a very fundamental question about a phenomenon that we call *implicit curricula*. Are examples

---

[*]Work performed while Xiaoxia Wu was student at UT Austin and interning at Blueshift.

[1]Code at https://github.com/google-research/understanding-curricula

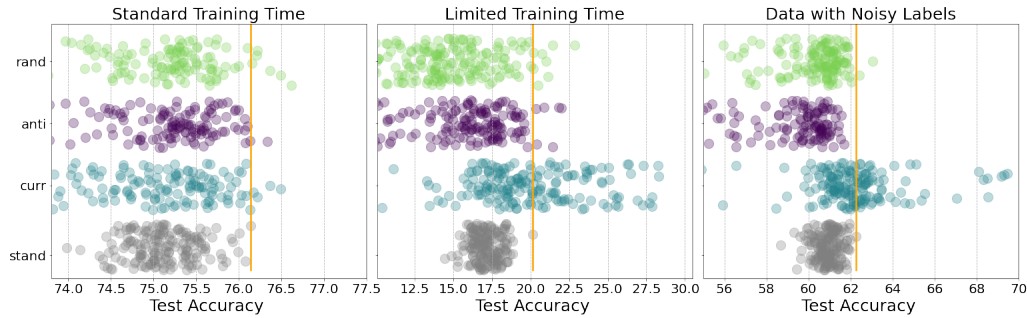

Figure 1: **Curricula help for time limited or noisy training, but not standard training.** Each point represents an independent learning ordering on CIFAR100 and is a mean over three independent runs with the same hyper-parameters. Color represents the type of learning, from bottom to top, are standard i.i.d. training (grey), curriculum (blue), anti-curriculum (purple), and random curriculum (green). The solid orange line is the best test accuracy for standard i.i.d. training. The left, middle and right plots represent standard-time, short-time, and noisy training. We find that for the original dataset and learning constraints there are no statistically significant benefits from anti, random, or curriculum learning (left). We find that for training with a limited time budget (center) or with noisy data (right) curriculum learning can be beneficial.

learned in a consistent order across different runs, architectures, and tasks? If such a robust notion exists, is it possible to change the order in which the examples are learned by presenting them in a different order? The answer to this question determines if there exists a robust notion of example *difficulty* that could be used to influence training.

We then look into different ways of associating *difficulty* to examples using *scoring functions* and a variety of schedules known as *pacing functions* for introducing examples to the training procedure. We investigate if any of these choices can improve over the standard full-data i.i.d. training procedure commonly used in machine learning. Inspired by the success of CL in large scale training scenarios, we train in settings intended to emulate these large scale settings. In particular, we study the effect of curricula when training with a training time budget and training in the presence of noise.

**Contributions.** In this paper, we systematically design and run extensive experiments to gain a better understanding of curricula. We train over 25,000 models over four datasets, CIFAR10/100, FOOD101, and FOOD101N covering a wide range of choices in designing curricula and arrive at the following conclusions:

- **Implicit Curricula: Examples are learned in a consistent order (Section 2).** We show that the order in which examples are learned is consistent across runs, similar training methods, and similar architectures. Furthermore, we show that it is possible to change this order by changing the order in which examples are presented during training. Finally, we establish that well-known notions of sample difficulty are highly correlated with each other.

- **Curricula achieve (almost) no improvement in the standard setting (Section 4 and 6).** We show curriculum learning, random, and anti-curriculum learning perform almost equally well in the standard setting.[2]

- **Curriculum learning improves over standard training when training time is limited (Section 5 and 6).** Imitating the large data regime, where training for multiple epochs is not feasible, we limit the number of iterations in the training algorithm and compare curriculum, random and anti-curriculum ordering against standard training. Our experiments reveal a clear advantage of curriculum learning over other methods.

- **Curriculum learning improves over standard training in noisy regime (Section 5 and 6).** Finally, we mimic noisy data by adding label noise to CIFAR100 and also use a natural noisy dataset – FOOD101N. Similar to Jiang et al. (2018); Saxena et al. (2019); Guo et al. (2018), our experiments indicate that curriculum learning has a clear advantage over other curricula and standard training.

**Related Work.** Bengio et al. (2009) is perhaps the most prominent work on curriculum learning where the "difficulty" of examples is determined by the loss value of a pre-trained model. Toneva

---

[2]See the first paragraph of Section B for details of the standard-time experimental setup.

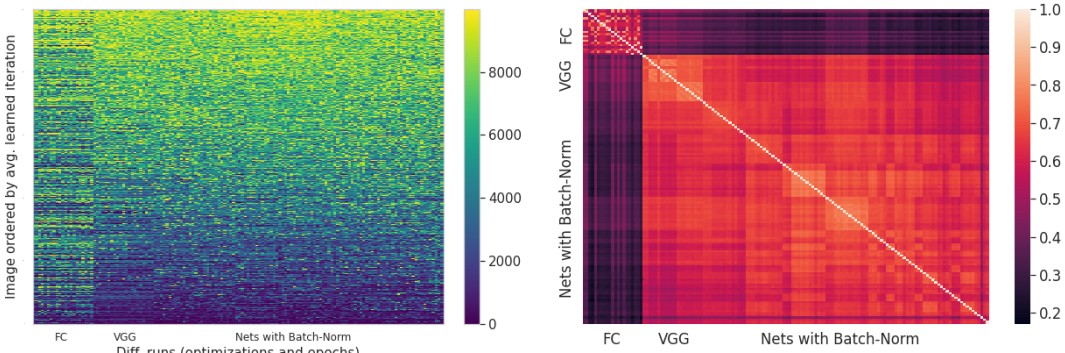

Figure 2: **Implicit Curricula: Images are learned in a similar order for similar architectures and training methods.** (Left) Epoch/Iteration at which each image is learned across 142 different architectures and optimization procedures. Each row is a CIFAR10 image ordered by its average learned epoch. The columns from left to right, are fully-connected (FC) nets, VGG nets (VGG11 and VGG19), and nets with Batch-Norm (Ioffe & Szegedy, 2015) including ResNet18, ResNet50, WideResNet28-10, WideResNet48-10 DenseNet121, EfficientNet B0, VGG11-BN and VGG19-BN. (Right) The Spearman correlation matrix shows high correlation between orderings within architecture families.

et al. (2019) instead suggested using the first iteration in which an example is learned and remains learned after that. Finally, Jiang et al. (2020b) has proposed using a consistency score (c-score) calculated based on the consistency of a model in correctly predicting a particular example's label trained on i.i.d. draws of the training set. When studying curriculum learning, we look into all of the above-suggested measures of sample difficulty. We further follow Hacohen & Weinshall (2019) in defining the notion of pacing function and use it to schedule how examples are introduced to the training procedure. However, we look into a much more comprehensive set of pacing functions and different tasks in this work. Please see Section A for a comprehensive review of the literature on curricula.

## 2 IMPLICIT CURRICULA

Curriculum learning is predicated on the expectation that we can adjust the course of learning by controlling the order of training examples. Despite the intuitive appeal, the connection between the order in which examples are shown to a network during training and the order in which a network learns to classify these examples correctly is not apriori obvious. To better understand this connection, we first study the order in which a network learns examples under traditional stochastic gradient descent with i.i.d. data sampling. We refer to this ordering – which results from the choice of architecture and optimization procedure – as an *implicit curriculum*.

To quantify this ordering we define the *learned iteration* of a sample for a given model as the epoch for which the model correctly predicts the sample for that and all subsequent epochs. Explicitly, $\min_{t^*}\{t^*|\hat{y}_\mathbf{w}(t)_i = y_i, \forall t^* \leq t \leq T\}$ where $y_i$ and $\hat{y}_\mathbf{w}(t)_i$ are the correct label and the predictions of the model for $i$-th data point (see the detailed mathematical description in Section 3.1).

We study a wide range of model families including fully-connected networks, VGG (Simonyan & Zisserman, 2014), ResNet (He et al., 2016), Wide-ResNet (Zagoruyko & Komodakis, 2016), DenseNet (Huang et al., 2017) and EfficientNet (Tan & Le, 2019) models with different optimization algorithms such as Adam (Kingma & Ba, 2014) and SGD with momentum (see Section B for details). The results in Figure 2 for CIFAR10 (Krizhevsky & Hinton, 2009) show that the implicit curricula are broadly consistent within model families. In particular, the ordering in which images are learned within convolutional networks is much more consistent than between convolutional networks and fully connected networks,[3] and the learned ordering within each sub-type of CNN is even more uniform. The robustness of this ordering, at least within model types, allows us to talk with less ambiguity about the difficulty of a given image without worrying that the notion of difficulty is highly model-dependent.

---

[3] The difference between shallow fully connected nets and deep convolutional nets, to some extent, matches what Mangalam & Prabhu (2019) has found when comparing shallow and deep networks.

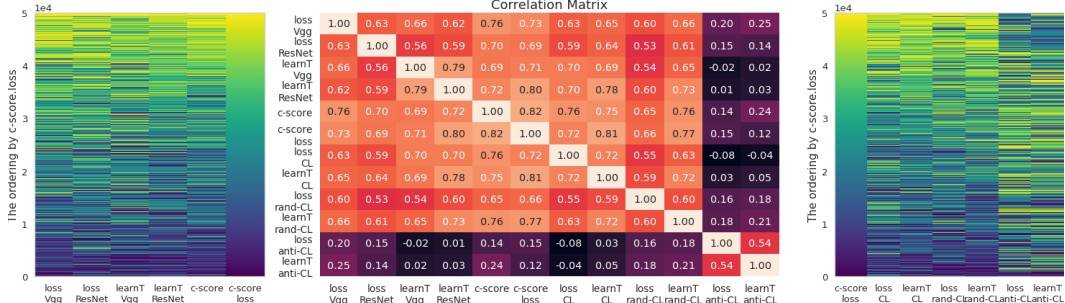

Figure 3: **Scoring functions show a high correlation for the standard training, but perceived difficulty depends on the training order.** (Left) Six scoring functions computed on CIFAR10 using the standard i.i.d. training algorithms. Columns from left to right show VGG-11 loss, ResNet-18 loss, VGG-11 iteration learned, ResNet-18 iteration learned, c-score, and a loss based c-score. Here the order given by VGG-11 loss or ResNet-18 loss uses the recorded loss at epoch 10. (Right) Loss-based difficulty and learned iteration difficulty when performing non-standard training from left to right columns are: c-score baseline, curriculum-based training, random-training, and anti-curriculum training. The last three used the same pacing function – step with $a = 0.8$ and $b = 0.2$. (Center) The Spearman's rank correlation is high between all scoring functions computed from ordinary or curriculum training, but lower for random training and significantly lower for anti-curriculum, indicating that the three orderings lead to networks learning samples in different orders.

We will see in the next section (and Figure 3) that, as expected, the choice of *explicit curriculum* can alter the order in which a network learns examples. The most dramatic manifestation of this is anti-curriculum learning where showing the network images in the reverse order indeed causes the network to learn more difficult images first. Next, we introduce the class of curricula we will consider for the remainder of the paper.

## 3 PUTTING CURRICULA THROUGH THEIR PACES

Many different approaches have been taken to implement curricula in machine learning. Here we focus on a particular widely used paradigm introduced in Bengio et al. (2009) and used in Hacohen & Weinshall (2019). In this setup, a curriculum is defined by specifying three ingredients,

- **The scoring function**: The scoring function is a map from an input example, $x$, to a numerical score, $s(x) \in \mathbb{R}$. This score is typically intended to correspond to a notion of difficulty, where a higher score corresponds to a more difficult example.

- **The pacing function**: The pacing function $g(t)$ specifies the size of the training data-set used at each step, $t$. The training set at step $t$ consists of the $g(t)$ lowest scored examples. Training batches are then sampled uniformly from this set.

- **The order**: Additionally we specify an order of either *curriculum* – ordering examples from lowest score to highest score, *anti-curriculum* – ordering examples from highest score to lowest, or *random*. Though technically redundant with redefining the scoring function, we maintain the convention that the score is ordered from easiest to hardest.

This procedure is summarized in Algorithm 1. It is worth emphasizing that due to the pacing function, using a random ordering in Algorithm 1 is not the same as traditional i.i.d. training on the full training dataset, but rather corresponds to i.i.d. training on a training dataset with dynamic size. We

---

**Algorithm 1** (Random-/Anti-) Curriculum learning with pacing and scoring functions

1: **Input:** Initial weights $\mathbf{w}^0$, training set $\{\mathbf{x}_1, \ldots, \mathbf{x}_N\}$, pacing function $g : [T] \to [N]$, scoring function $s : [N] \to \mathbb{R}$, order $o \in \{$"ascending", "descending", "random"$\}$.
2: $(\mathbf{x}_1, \ldots, \mathbf{x}_N) \leftarrow \text{sort}(\{\mathbf{x}_1, \ldots, \mathbf{x}_N\}, s, o)$
3: **for** $t = 1, \ldots, T$ **do**
4: $\quad \mathbf{w}^{(t)} \leftarrow \text{train-one-epoch}(\mathbf{w}^{(t-1)}, \{\mathbf{x}_1, \ldots, \mathbf{x}_{g(t)}\})$
5: **end for**

---

stress that the scoring and pacing function paradigm for curriculum learning is inherently limited. In this setup, the scoring function is computed before training over all of the data and thus the algorithm cannot implement a self-paced and training-dependent curriculum as has been considered in Kumar et al. (2010); Jiang et al. (2015); Platanios et al. (2019). Additionally, the dynamic training dataset is built by including all examples within a fixed score window (from lowest score up in curricula and highest score down in anti-curricula) and does not accommodate more flexible subsets. Furthermore, the form of curriculum discussed here only involves ordering examples from a fixed training dataset, rather than more drastic modifications of the training procedure, such as gradually increasing image resolution (Vogelsang et al., 2018) or the classes (Weinshall et al., 2018). Nonetheless, it is commonly studied and serves as a useful framework and control study to empirically investigate the relative benefits of training orderings.

Next, we describe scoring and pacing functions that will be used in our empirical investigation.

## 3.1 SCORING FUNCTIONS: DIFFICULTY SCORES ARE BROADLY CONSISTENT

In this section, we investigate three families of scoring functions. As discussed above, we define the scoring function $s(\boldsymbol{x}, y) \in \mathbb{R}$ to return a measure of an example's difficulty. We say that an example $\boldsymbol{x}_j$ is more difficult than an example $\boldsymbol{x}_i$ if $s(\boldsymbol{x}_j, y_j) > s(\boldsymbol{x}_i, y_i)$. In this work, we consider three scoring functions:

- **Loss function.** In this case samples are scored using the real-valued loss of a reference model that is trained on the same training data,e.g. given a trained model $f_{\mathbf{w}} : \mathcal{X} \to \mathcal{Y}$, we set $s(\boldsymbol{x}_i, y_i) = \ell(f_{\mathbf{w}}(\boldsymbol{x}_i), y_i)$.

- **Learned epoch/iteration.** This metric has been introduced in Section 2. We let $s(\boldsymbol{x}_i, y_i) = \min_{t^*} \{t^* | \hat{y}_{\mathbf{w}}(t)_i = y_i, \forall t^* \leq t \leq T\}$ (see Algorithm 3 and Figure 11 for an example).

- **Estimated c-score.** c-score (Jiang et al., 2020b) is designed to capture the consistency of a reference model correctly predicting a particular example's label when trained on independent i.i.d. draws of a fixed sized dataset not containing that example. Formally, $s(\boldsymbol{x}_i, y_i) = -\mathbb{E}_{D \stackrel{n}{\sim} \hat{\mathcal{D}} \setminus \{(\boldsymbol{x}_i, y_i)\}} [\mathbb{P}(\hat{y}_{\mathbf{w},i} = y_i | D)]$ where $D$, with $|D| = n$, is a training data set sampled from the data pool without the instance $(\boldsymbol{x}_i, y_i)$ and $\hat{y}_{\mathbf{w},i}$ is the reference model prediction. We also consider a loss based c-score, $s(\boldsymbol{x}_i, y_i) = \mathbb{E}^r_{D \stackrel{n}{\sim} \hat{\mathcal{D}} \setminus \{(\boldsymbol{x}_i, y_i)\}} [\ell(\boldsymbol{x}_i, y_i) | D]$, where $\ell$ is the loss function. The pseudo-code is described in Algorithm 4.

To better understand these three scoring functions, we evaluate their consistency in CIFAR10 images. In particular, for VGG-11 and ResNet-18, we averaged the learned epoch over five random seeds and recorded the loss at epochs $2, 10, 30, 60, 90, 120, 150, 180, 200$. We also compute the c-score using multiple independently trained ResNet-18 models and compare it to the reference c-scores reported in (Jiang et al., 2020b). The main results of these evaluations are reported in the left and middle panels of Figure 3. We see that all six scores presented have high Spearman's rank correlation suggesting a consistent notion of difficulty across these three scores and two models. For this reason, we use only the c-score scoring function in the remainder of this paper.

Additional model seeds and training times are presented in Figure 12 and 13 in the appendix. Among all cases, we found one notable outlier. For ResNet-18 after $180$ epochs, the training loss no longer correlates with the other scoring functions. We speculate that this is perhaps due to the model memorizing the training data and achieving near-zero loss on all training images.

## 3.2 PACING FUNCTIONS: FORCING EXPLICIT CURRICULA

The pacing function determines the size of training data to be used at iteration $t$ (see Algorithm 1). We consider six function families: logarithmic, exponential, step, linear, quadratic, and root. Table 2 illustrates the pacing functions used for our experiments which is parameterized by $(a, b)$. Here $a$ is the fraction of training needed for the pacing function to reach the size of the full training set, and $b$ is the fraction of the training set used at the start of training, thus any pacing function with $a = 0$ or $b = 1$ is equivalent to standard training. We denote the full training set size by $N$ and the total number of training steps by $T$. Explicit expressions for the pacing functions we use, $g_{(a,b)}(t)$, and examples are shown in Figure 4.

In order to cover many possible choices of pacing function, in all remaining experiments, we select $b \in \{0.0025, 0.1, 0.2, 0.4, 0.8\}$ and $a \in \{0.01, 0.1, 0.2, 0.4, 0.8, 1.6\}$ for our empirical study. Pacing functions with these parameters are plotted in Figure 14.

Given these pacing functions, we now ask if the explicit curricula enforced by them can change the order in which examples are learned. The right panel in Figure 3 indicates what is the impact of using curriculum, anti-curriculum or random ordering on the order in which examples are learned. All curricula use the step pacing functions with $a = 0.8$ and $b = 0.2$. The result indicates that curriculum and random ordering do not change the order compared to c-score but anti-curriculum could indeed force the model to learn more difficult examples sooner. This is also demonstrated clearly in the Spearman's correlation of anti-curriculum with other methods shown in the middle panel of Figure 3.

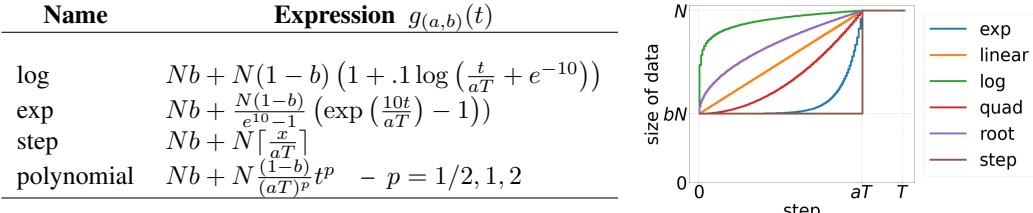

Figure 4: **Pacing functions** (Left) pacing function definitions for the six families of pacing functions used throughout. (Right) example, pacing function curves from each family. The parameter $a$ determines the fraction of training time until all data is used. The parameter $b$ sets the initial fraction of the data used.

## 4  PACING FUNCTIONS GIVE MARGINAL BENEFIT, CURRICULA GIVE NONE

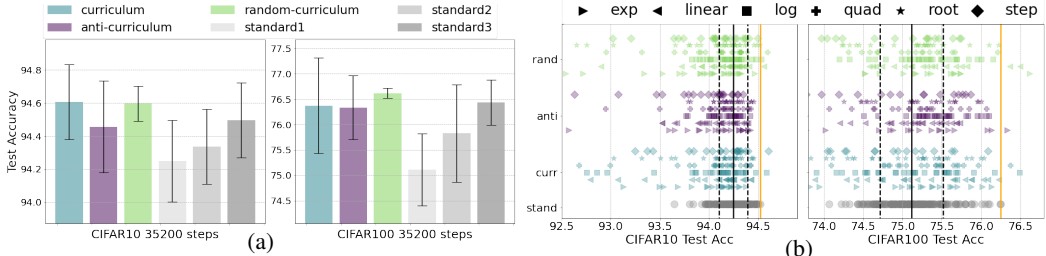

Figure 5: **Curricula provide little benefit for standard learning.** (a) Bar plots showing the best mean accuracy, for curriculum (blue), anti-curriculum (purple), random-curriculum (green), and standard i.i.d. training (grey) with three ways of calculating the standard training baseline for CIFAR10 (left) and CIFAR100 (right). (b) Means over three seeds for all 540 strategies, and 180 means over three standard training runs (grey) for CIFAR10 (left) and CIFAR100 (right). The x-axis is the test accuracy. Marker shape signifies the pacing function family. Black solid and dashed lines give the mean and standard deviation over standard training runs. The solid orange line is the standard2 baseline test accuracy. We observe no statistically significant improvement from curriculum, anti-curriculum, or random training.

Equipped with the ingredients described in the previous sections, we investigated the relative benefits of (random-/anti-) curriculum learning. As discussed above, we used the c-score as our scoring function and performed a hyper-parameter sweep over the 180 pacing functions and three orderings described in Section 3. We replicated this procedure over three random seeds and the standard benchmark datasets, CIFAR10 and CIFAR100. For each experiment in this sweep, we trained a ResNet-50 model for 35200 total training steps[4]. We selected the best configuration and stopping time using the validation accuracy and recorded the corresponding test accuracy. The results of these runs for CIFAR10/100 are shown in Figure 5.

To understand whether (anti-) curriculum, or random learning provides any benefit over ordinary training, we ran 540 standard training runs. From these 540 runs, we created three baselines. The *standard1* baseline is the mean performance over all 540 runs. The *standard2* baseline splits the 540

---

[4]This step number corresponds to 100 epochs of standard training (with batch size 128).

Figure 6: **Curriculum-learning helps when training with a limited time budget.** CIFAR100 performance when training with 17600, 1760 and 352 total steps (see Figure 17 for CIFAR10). Curriculum learning provides a robust benefit when training for 1760 and 352 steps. See Figure 5 for additional plotting details.

runs into 180 groups of 3; the mean is taken over each group of 3 and the maximum is taken over the 180 means. This setup is intended as a stand in for the hyperparameter search of the 30 choices of pacing parameters $(a, b)$ each with three seeds. Lastly, *standard3* is the mean value of the top three values over all 540 runs.

**Marginal value of ordered learning.** When comparing our full search of pacing functions to the naive mean, *standard1*, all three sample orderings have many pacing functions that outperform the baseline. We find, however, that this is consistent with being an artifact of the large search space. When comparing the performance of all three methods to a less crippled baseline, which considers the massive hyperparameter sweep, we find that none of the pacing functions or orderings statistically significantly outperforms additional sampling of ordinary SGD training runs. Furthermore, we establish that using similar techniques to remove examples from the training set (as opposed to introducing them) also does not help (see Figure 16 in the appendix).

Perhaps most striking is that performance shows no dependence on the three different orderings (and thus scoring function). For example, in the CIFAR10 runs, the best mean accuracy is achieved via random ordering, while in CIFAR100, the best single run has a random ordering. This suggests that for these benchmark datasets, with standard training limitations, any marginal benefit of ordered learning is due entirely to the dynamic dataset size (pacing function).

## 5 CURRICULA FOR SHORT-TIME TRAINING AND NOISY DATA

In the previous section, we found little evidence for statistically significant benefits from curricula or pacing. This observation is consistent with the fact that curricula have not become a standard part of supervised image classification. In other contexts, however, curricula are standard. Notably, in practice, many large scale text models are trained using curricula (Brown et al., 2020; Raffel et al., 2019). These models are typically trained in a data-rich setting where multiple epochs of training is not feasible. Furthermore, the data for training is far less clean than standard image benchmarks.

To emulate these characteristics and investigate whether curricula are beneficial in such contexts, we applied the same empirical analysis used in Section 4 to training with a limited computational budget and training on noisy data. It should be noted that the benefits of curricula for noisy data have been studied previously, for example, in (Jiang et al., 2018; Saxena et al., 2019; Guo et al., 2018).

**Limited training time budget.** For shorter time training, we follow the same experiment setup described in Section 4 but modify the total number of steps, $T$. Instead of using 35200 steps, we consider training with $T = 352, 1760$ or 17600 steps. We use cosine learning rate decay, decreasing monotonically to zero over the $T$ steps. The results are given in Figure 6, and in supplementary Figures 20 and 21 for CIFAR10/100.

We see that curricula are still no better than standard training for 17600 iterations. However, when drastically limiting the training time to 1760 or 352 steps, curriculum-learning, but not anti-curriculum can indeed improve performance, with increasing performance gains as the the training time budget is decreased. Surprisingly, at the smallest time budget, 352 steps, we also see random-learning consistently outperforms the baseline, suggesting that the dynamic pacing function on its own helps improve performance. We depict the best pacing functions for 1760 and 352 steps in Figure 8. In the 2nd plot of Fig 8 (left, limit training), many pacing functions start with a relatively small dataset size and maintain a small dataset for a large fraction of the total training time.

**Data with noisy labels.** To study curricula in the context of noisy data, we adopted a common procedure of generating artificial label noise by randomly permuting the labels (Zhang et al., 2017;

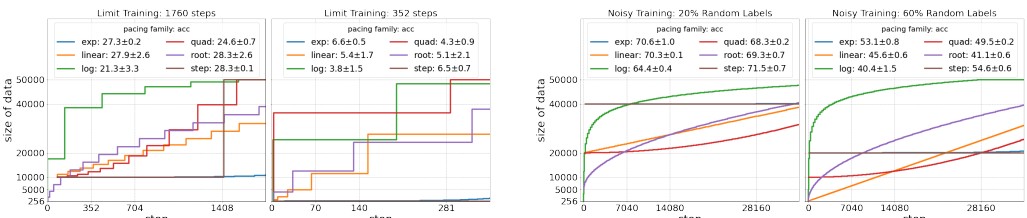

Figure 7: **Curriculum-learning helps when training with noisy labels.** Performance on CIFAR100 with the addition of 20%, 40%, 60% and 80% label noise shows robust benefits when using curricula.

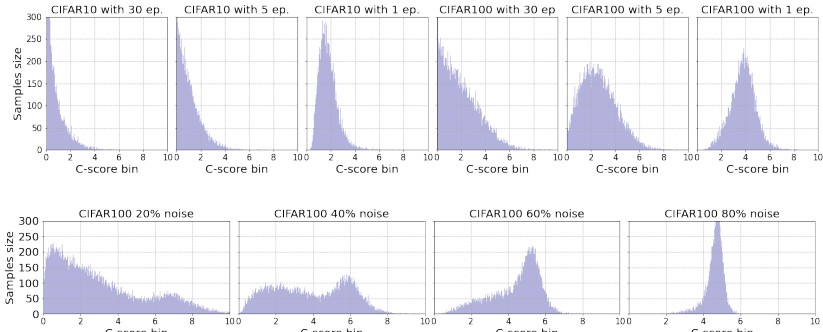

Figure 8: **Top performing pacing functions for limited time and noisy training.** Top performing pacing functions from the six families considered for CIFAR100 (from left to right) finite time training with 1760 and 352 steps, 20% label noise and 60% label noise.

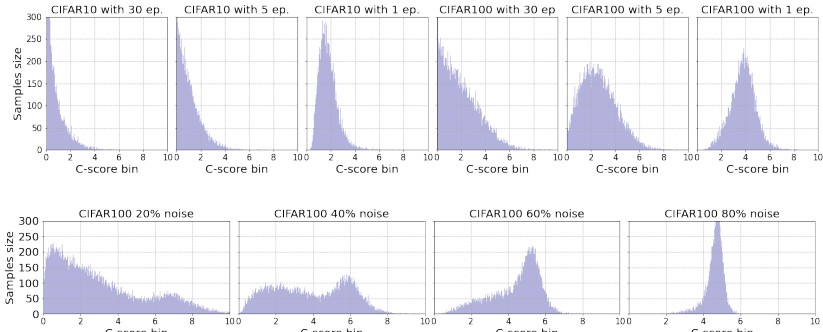

Figure 9: **Label noise and limited training time shift the c-score distributions towards more difficult examples.** We compute loss-based c-score for limited time training on CIFAR10 (top left three), CIFAR100 (top right three) and for CIFAR100 models trained with label noise (bottom row).

Saxena et al., 2019; Jiang et al., 2020b). We considered CIFAR100 datasets with 20%, 40%, 60%, and 80% label noise and otherwise use the same experimental setup described in Section 4. As the training data has been modified, we must recompute the c-score. The results are shown in Figure 9.

Equipped with the new ordering, we repeat the same set of experiments and obtain the results shown in Figure 7 and supplementary Figure 22. Figure 7 shows that curriculum learning outperforms other methods by a large margin across all noise levels considered. The best pacing function in each family is shown in Figure 8. We see that the best overall pacing functions for both 20% and 60% noise are the step and exponential pacing functions corresponding to simply ignoring all noisy data during training.For 40% noisy labels, this strategy was not contained in our sweep of $a$ and $b$ values.

To understand why the reduced training time and label noise benefit from curricula, we plot in Figure 9 the loss-based c-score distributions for CIFAR10/100 trained with 30, 5 and 1 epochs and CIFAR100 trained with noise 20%, 40%, 60%, and 80%. For the distribution of clean CIFAR100/10 with 30 epochs (see title), a significant number of images concentrate around zero. However, the concentration slowly shifts to larger values as the the total training epoch decreases or the label noise increases. This suggests that both label noise and reduced time training induce a more difficult c-score distribution and curricula can help by focusing first on the easier examples.

## 6 CURRICULA IN THE LARGE DATA REGIME

In this section, we are interested in investigating whether the previous conclusions we made for CIFAR10/100 generalize to larger scale datasets. We conduct experiments on FOOD101 (Bossard

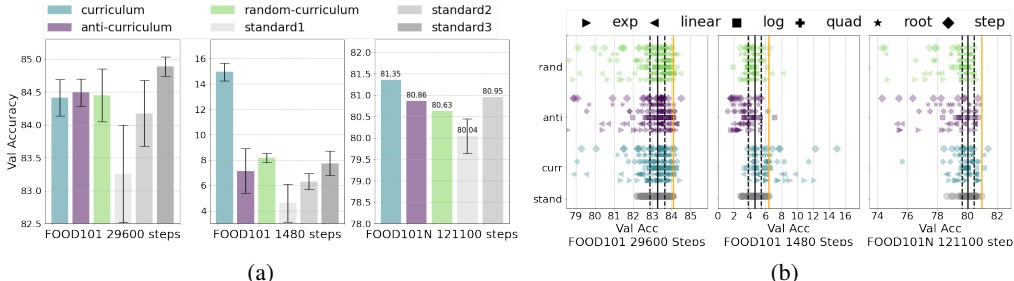

Figure 10: **Large data regime – FOOD101 and FOOD101N.** (a) Bar plots showing the best mean accuracy, for curriculum (blue), anti-curriculum (purple), random-curriculum (green), and standard i.i.d. training (grey) with three ways of calculating the standard training baseline for FOOD101 standard-time training (left), FOOD101 short-time training (middle), and FOOD101N training (right)[4]. (b) For FOOD101 standard-time training (left), we plot the means over three seeds for all 540 strategies, and 180 means over three standard training runs (grey). For FOOD101 short-time training (middle), we plot the means over three seeds for all 216 strategies, and 72 means over three standard training runs. For FOOD101N (right), we reported the values for all 216 strategies and 72 standard training runs. See Figure 5 for additional details. In FOOD101 standard-time training, we observe no statistically significant improvement from curriculum, anti-curriculum, or random training. However, for FOOD101 short-time training and FOOD101N training, CL provides a robust benefit.

[4] Since we only run one random seed for FOOD101N, we plot the average of all standard runs (standard1) and the best run over 72 strategies (standard2).

et al., 2014) which consists of $75,000$ training and $25,000$ validation images and FOOD101N (Lee et al., 2018) which consists $310,000$ training images and the same validation images as FOOD101. The maximum length of the image dimensions are 512 pixels. We train the ResNet-50 model by resizing these images to be 250 and applying random horizontal flips and random crops with size 224 as in the PyTorch example for ImageNet. See Section B for more experiment details.

**Standard Training Time.** For standard-time training, the total number of training steps is $296 \times 100$, which corresponds to 100 epochs of standard i.i.d. training. Similar to Section 4, we train $180 \times 3$ models respectively for standard i.i.d., curriculum, anti-curriculum and random-curriculum learning over three random seeds. The result is presented in Figure 10. Again, we observe none of the orderings or pacing functions significantly outperforms standard i.i.d. SGD learning.

**Limited time Training and Noisy Labels Training.** For FOOD101 short-time training, models are trained for $296 \times 5$ steps. We do not search over 180 pacing functions. Instead, we consider 72 pacing functions with $a \in \{0.1, 0.4, 0.8, 1.6\}$ and $b \in \{0.1, 0.4, 0.8\}$ over the six pacing function families. For FOOD101N training, models are trained for $121,100$ steps with a single fixed random seed 111 and the same reduced pacing function search space as FOOD101. Although the hyper-parameter search space for both FOOD101 and FOOD101N training is smaller than the one used in Section 5, the conclusion is similar – Figure 10 shows curriculum still perform better than anti-curriculum, random curriculum and standard SGD training.

In summary, the observations on CIFAR10 and CIFAR100 in three training regime – standard-time, short-time and noisy training – generalize to larger-scale data like FOOD101 and FOOD101N.

## DISCUSSION

In this work, we established the phenomena of *implicit curricula*, which suggests that examples are learned in a consistent order. We further ran extensive experiments over four datasets, CIFAR10, CIFAR100, FOOD101 and FOOD101N and showed that while curricula are not helpful in standard training settings, easy-to-difficult ordering can be beneficial when training with a limited time budget or noisy labels.

We acknowledge that despite training more than 25,000 models on CIFAR-like and FOOD101(N) datasets, our empirical investigation is still limited by the computing budget. That limitation forced us to choose between diversity in the choice of orderings and pacing functions as opposed to diversity in the tasks type, task scale, and model scale. Our strategic choice of limiting the datasets to CIFAR-10/100 and FOOD-101(N) allowed us to focus on a wide range of curricula and pacing functions, which was very informative. As a result, our general statements are more robust with respect to the choice of curricula – which is the main object of study in this paper – but one should be careful about generalizing them to tasks/training regimes that are not similar to those included in this work.

ACKNOWLEDGEMENTS

We would like to thank for the discussions with the teammates at Google. We owe particular gratitude to Mike Mozer, Vaishnavh Nagarajan, Preetum Nakkiran, Hanie Sedghi, and Chiyuan Zhang. We appreciate the help with the experiments from Zachary Cain, Sam Ritchie, Ambrose Slone, and Vaibhav Singh. Lastly, XW would like to thank the team for all of the hospitality.

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

## A    Literature on Curriculum Learning

The related work can be divided into the following four main subparts. With the best of our efforts, we mainly cover the recent ones among the numerous papers.

**Understanding the learning dynamics of individual samples.** It is critical to understand the learning dynamics of a model and so bridge the gap between human learning and curriculum learning (Khan et al., 2011). An interesting phenomenon for the standard i.i.d. training algorithm is observed and termed "catastrophic forgetting" – neural networks cannot learn in a sequence manner and suffer a significant drop in generalization on earlier tasks (French, 1999; Goodfellow et al., 2014; Kirkpatrick et al., 2017; Ritter et al., 2018; Nguyen et al., 2019; Ramasesh et al., 2020). Even for a single task, forgetting events occur with high frequency to some examples, which could lie in the subset of "hard" samples (Toneva et al., 2019). To further understand how the forgettable examples are related to "hard" examples and unforgettable examples to "easy" ones, we select a commonly used metric, loss functions, to define the difficulty (Bengio et al., 2009; Hacohen & Weinshall, 2019). One criticism of using loss functions as a metric is that over-parameterized neural networks can memorize and completely overfit the training data reaching the interpretation region if trained long enough (Arpit et al., 2017; Zhang et al., 2017), which has triggered plenty of theoretical and empirical analysis (Chatterjee, 2018; Yun et al., 2019; Ge et al., 2019; Gu & Tresp, 2019; Zhang et al., 2020). Thus, in addition to loss functions, we measure the difficulty of examples with a consistency score (c-score) metric proposed in Jiang et al. (2020b), in which the model should never see the estimated instances, and the estimation should be accurate, close to full expectation.

**Improved methods of curriculum learning.** Fruitful methods have been proposed to improve scoring functions and pacing functions in various applications. One significant improvement is about dynamically updating the ordering and pacing based on the current models or a learned teacher model. This idea can be positively related to importance sampling methods for empirical risk minimization (Gretton et al., 2009; Shimodaira, 2000; Needell et al., 2014; Johnson & Guestrin, 2018; Byrd & Lipton, 2019; Khim et al., 2020). CL in the earlier or short-time training assigns the accessible instances with higher weight than difficult ones. Self-paced curriculum learning (Jiang et al., 2015; 2018), based on self-paced learning (Kumar et al., 2010), formulate a concise optimization problem that considers knowledge before and during training. Huang et al. (2020) adapts CL to define a new loss function such that the relative importance of easy and hard samples are adjusted during different training stages, and demonstrated its effectiveness on popular facial benchmarks. The curriculum of teacher's guidance (Graves et al., 2017) or teacher-student collaboration (Matiisen et al., 2019) have been empirically proven useful in reinforcement learning for addressing sequential decision tasks. A comprehensive survey on curriculum reinforcement learning can be found in Weng (2020) and Platanios et al. (2019). Our work is not about proposing a method to improve CL training but empirically exploring CL's performance to clear the pictures when curricula help and when not.

---

**Algorithm 2** Loss function

---

1: **Input:** Initial weights $\mathbf{w}^0$, training set $\{\mathbf{x}_1, \ldots, \mathbf{x}_N\}$
2: **for** $t = 1, \ldots, T$ **do**
3:    $\mathbf{w}^{(t)} \leftarrow$ train-one-epoch$(\mathbf{w}^{(t-1)}, \{\mathbf{x}_1, \ldots, \mathbf{x}_N\})$
4: **end for**
5: **for** $i = 1, \ldots, N$ **do**
6:    $s(i) \leftarrow \ell(f_{\mathbf{w}^{(T)}}(\mathbf{x}_i), y_i)$
7: **end for**

---

---

**Algorithm 3** Learned Epoch

---

1: **Input:** Initial weights $\mathbf{w}^0$, training set $\{\mathbf{x}_1, \ldots, \mathbf{x}_N\}$
2: **for** $i = 1, \ldots, N$ **do**
3:    $s(i) \leftarrow 0$
4:    $c(i) \leftarrow (T+1)\ell_{0/1}(f_{\mathbf{w}^{(0)}}(\mathbf{x}_i), y_i)$
5: **end for**
6: **for** $t = 1, \ldots, T$ **do**
7:    $\mathbf{w}^{(t)} \leftarrow$ train-one-epoch$(\mathbf{w}^{(t-1)}, \{\mathbf{x}_1, \ldots, \mathbf{x}_N\})$
8:    **for** $i = 1, \ldots, N$ **do**
9:       **if** $\ell_{0/1}(f_{\mathbf{w}^{(t)}}(\mathbf{x}_i), y_i) = 0$ **then**
10:         $c(i) \leftarrow \min(c(i), t)$
11:       **else**
12:         $c(i) \leftarrow T+1$
13:       **end if**
14:       $s(i) \leftarrow s(i) + \ell(f_{\mathbf{w}^{(t)}}(\mathbf{x}_i), y_i)/T$
15:    **end for**
16: **end for**
17: **for** $i = 1, \ldots, N$ **do**
18:    $s(i) \leftarrow s(i) + c(i)$
19: **end for**

---

**Curriculum Learning with label noise.** CL has been widely adopted with new methods to attain positive results in the learning region with artificial or natural label noises (Natarajan et al., 2013; Sukhbaatar & Fergus, 2014; Zhang et al., 2017; Jiang et al., 2020a). Guo et al. (2018) splits the data into three subsets according to noisy level and applies different weights for each subset during different training phases, which achieved the state-of-the-art result in the WebVision data set (Li et al., 2017). Ren et al. (2018) uses meta-learning to learn the weights for training examples based on their gradient directions. Saxena et al. (2019) obtains significant improvement by equipping each instance with a learnable parameter governing their importance in the learning process. Shen & Sanghavi (2019) and Shah et al. (2020) suggest iteratively discards those high current losses to make the optimization robust to noisy labels. Lyu & Tsang (2020) presents a (noise pruned) curriculum loss function with theoretical guarantees and empirical validation for robust learning. Given these success cases, empirical investigation with label noise for CL is within our scope.

**Anti-curriculum v.s. large current loss minimization.** Although both anti-curriculum learning and large current loss minimization focus on "difficult" examples, the two ideas could be orthogonal, particularly in longer training time. The former relies on a static pre-defined ordering while the latter requires the order iteratively updated. For a short time learning, the two could be similar. Among much work on minimizing large current loss (i.e., Fan et al. (2017); Zhang et al. (2019a); Jiang et al. (2019); Ortego et al. (2019) and reference therein), Kawaguchi & Lu (2020) introduced ordered stochastic gradient descent (SGD), which purposely biased toward those instances with higher current losses. They empirically show that the ordered SGD outperforms the standard SGD (Bottou et al., 2018) when decreasing the learning rate, even though it is not better initially. Supported with big loss minimization (short-time training), hard data-mining methods, and some empirical success of anti-curriculum (mentioned in the introduction), we investigate anti-curriculum.

---

**Algorithm 4** Estimated c-score

1: **Input:** training set $\mathcal{S} = \{\mathbf{x}_1, \ldots, \mathbf{x}_N\}$, ratio $\alpha$, number of repeated experiments $r$
2: **for** $k = 1, 2, \ldots r$ **do**
3:    **for** $j = 1, \ldots \lceil 1/\alpha \rceil$ **do**
4:       $\mathcal{S}_{\text{test}} = \left\{ \mathbf{x}_{\lceil (j-1)\alpha N \rceil + 1}, \ldots, \mathbf{x}_{\lceil j\alpha N \rceil} \right\}$
5:       $\mathbf{w} \leftarrow \text{train}(\mathcal{S}/\mathcal{S}_{\text{test}})$
6:       **for** $\mathbf{x}_i \in \mathcal{S}_{\text{test}}$ **do**
7:          $s(i) \leftarrow s(i) + \ell(f_{\mathbf{w}}(\mathbf{x}_i), y_i)/r$
8:       **end for**
9:    **end for**
10: **end for**

---

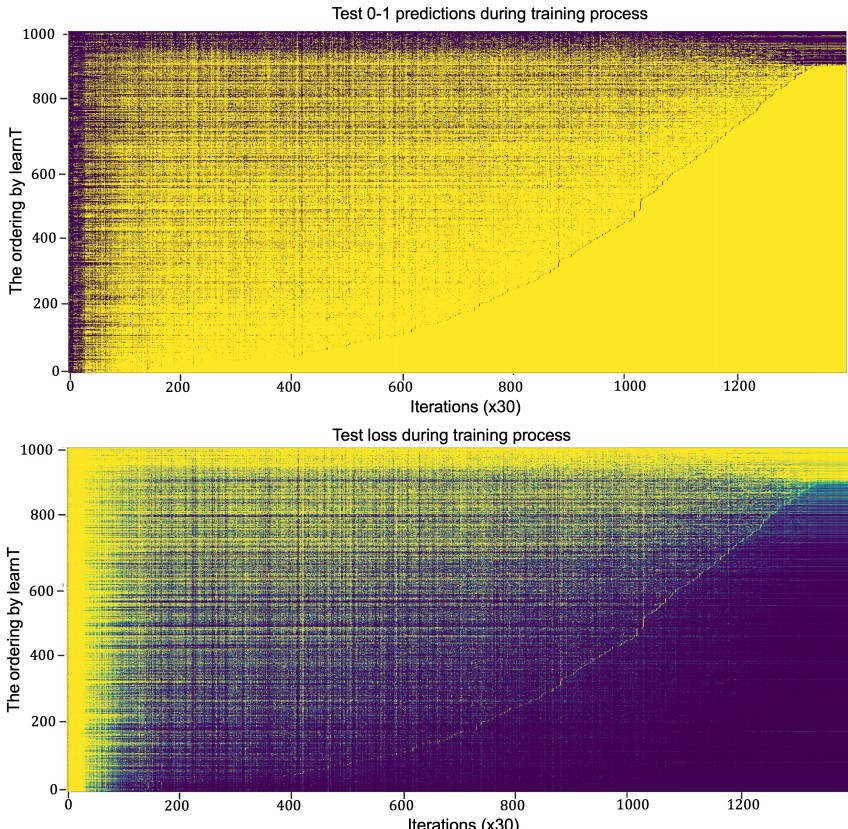

Figure 11: The top figure plots the 0 (dark blue) and 1 (yellow) prediction for the images (y-axis) during the iterations (x-axis). The bottom figure plots the corresponding loss values. The images belong to class 9 in the testing set of CIFAR10. We use the standard time training with batch-size 256 (Section B) and record the loss and 0-1 predication every 30 iterations.

## B   EXPERIMENT DETAILS

We summarize the common experimental setup first and give details for each figure. We first clarify *standard training* that appears in most of the figures (such as Figure 1, 5 and 7), where we are based on the code in PyTorch and use ResNet50 model with default initialization.[5] The data augmentation includes random horizontal flip and normalization, and the random training seeds are fixed to be $\{111, 222, 333\}$. We choose a batch size to be 128 and use one NVIDIA Tesla V100 GPU for each experiment. We use Caliban (Ritchie et al., 2020) and Google cloud AI platform to submit the jobs. The optimizer is SGD with 0.9 momentum, weight decay $5 \times 10^{-5}$, and a learning rate scheduler -

---

[5]https://github.com/pytorch/examples/blob/master/imagenet/main.py

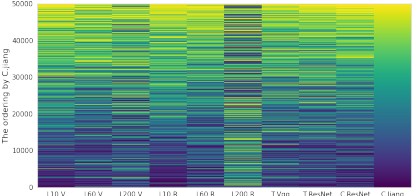

Figure 12: The left plot is the visualization of the ten orderings of CIFAR10 using the standard i.i.d. training algorithms (a) *L10.V*, *L60.V* and *L200.V*: the loss function of VGG11 at epoch 10, 60 and 200. (b) *L10.R*: the loss function by ResNet18 at epoch 10, 60 and 200. (c) *T.Vgg*: the learned iteration by VGG-11 (d) *learnT.ResNet*: the learned iteration by ResNet18 (f) *C.ResNet*: the c-score by ResNet18 with much less computation than (e). The Spearman correlation matrix for the orderings is plotted on the right. Note that we select *L10.V*, *L10.R* and the last four ordering to present in Figure 3.

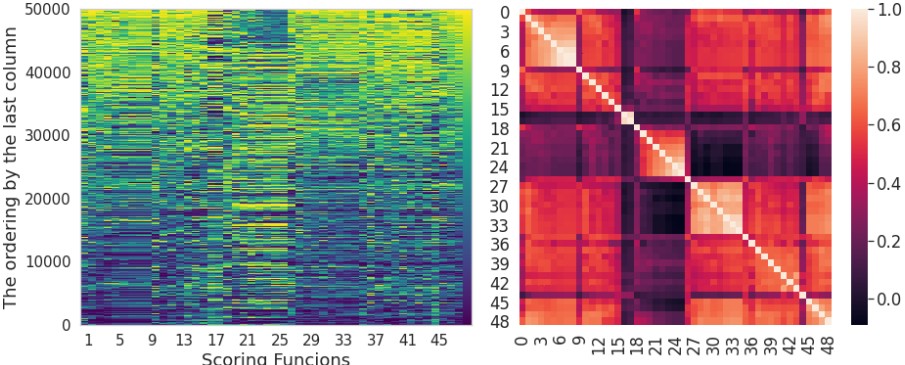

Figure 13: The left plot is the visualization of the ten orderings of CIFAR10 using the standard i.i.d. training algorithms, curriculum, anti-curriculum and random-curriculum learning. All columns but the last four are using the loss function as scoring metric (a) 1-9 columns are VGG11 at epoch $2, 10, 30, 60, 90, 120, 150, 180, 200$. (b) 10-18 columns are ResNet18 at epoch $2, 10, 30, 60, 90, 120, 150, 180, 200$. (c) 19-27, 28-36, 37-45 columns are anti-curriclum, curriculum and random curriculum learning with step-pacing function parameter $a = 1.0$ and $b = 0.2$ at step 2291, 7031, 11771, 16511, 21251, 25991, 28361, 30731 and 39100 (d) 46-49 are *learnT.Vgg*, *learnT.ResNet*, *c-score.loss* and *c-score* as described in Figure 3 and Figure 12. The right plot is the corresponding Spearman correlation matrix.

cosine decay – with an initial value of $0.1$. We use half-precision for training. In most of our cases, when using "standard training setup", we will implicitly mean the total training epoch is $100$. For the figures in Section 2 and 3, we use the full training samples $50000$. For figures in Section 4 and 5, we use training samples $45000$ and validation samples $5000$. We look for the best test error of these $5000$ validation samples and plot the corresponding test error/prediction.

For other figures such as Figure 3, 12 and 13. We vary the optimizers by either completely switching to Adam, or using different learning rate schedulers such as step-size decay every 30 epochs. The batch size are chosen to be $128$ and $256$, and weight decay $5 \times 10^{-5}$, $1 \times 10^{-5}$, $1 \times 10^{-6}$, or $0$.

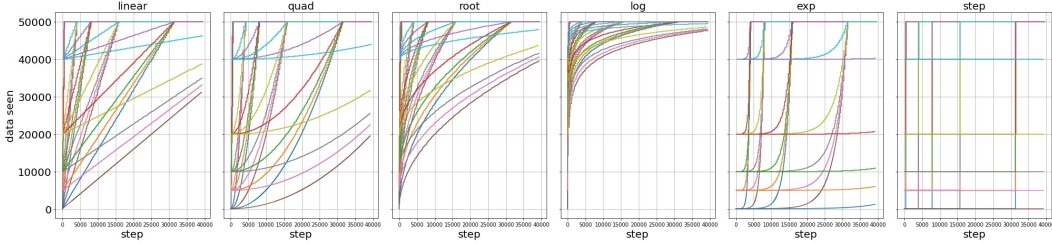

Figure 14: Visualization of the six types of pacing functions for $T = 39100$ and $N = 50000$ with $a \in \{0.01, 0.1, 0.2, 0.4, 1.0, 1.6\}$ and $b \in \{0.0025, 0.1, 0.2, 0.4, 1.0\}$.

When using fully-connected nets, VGG, and other networks, we pick the learning rates $0.01$ among grid search in $\{0.5, 0.1, 0.01, 0.001\}$.

For (anti-/random-) curriculum learning, we move the $torch.utils.data.DataLoader$ in the training loop so that to update the size of training data dynamically. We also refer to the code in Hacohen & Weinshall (2019).[6] In particular, each time we load data, we make class balance a priority, which means we do not completely follow the exact ordered examples. However, our ordering does imply the ordering within a class.

In the following subparts, we give details for each figure.

**Figure 2 in Section 3.** Algorithm 3 illustrate the main steps we take to measure the order. We now present the configuration for each experiment that translates to the ordering. We design three layers for the architecture in fully connected networks (with hidden unites $5120$ for the first layer and $500$ or $1000$ for the second one). ReLU activation function is used between the layers. For VGG networks, we use VGG11 and VGG19. For both networks, we use a learning rate of $0.01$ and weight-decay $0.0005$. We vary the random seeds and the optimizers, as mentioned above. For the Adam optimizer, we set epoch to be $100$ and initialize the learning rate $0.001$ decaying by a factor of $10$ after $50$ epochs. For SGD with $0.9$ momentum ($100$ epochs), we use cosine learning rate schedules or a decaying schedule with a factor of $10$ after $100//3$ epochs. Moreover, we widen the choice of convolutional networks including VGG11(19)-BN denoted by VGG11(19) with batch normalization, ResNet18, ResNet50, WideResNet28-10, WideResNet48-10 DenseNet121, EfficientNet B0. We set the learning rate for standard SGD training – $0.1$. Finally, we vary the training epochs and choose a longer one, $200$ epochs, to check the consistency. In summary, we present all the experiments in Figure 15 where we have $seed\_\# - arc\_\# - scheduler\_\#$ with $\#$ indicating the configuration for SGD with $0.9$ momentum training, and $seed\_\# - arc\_\# - adam$ for Adam. We add $.1$ to indicate those longer $200$-epoch training.

**Figure 3, 12 and 13.** We mainly focus on filling the details on how to train the models since Algorithm 2 and 4 are straightforward to implement. For VGG11 and ResNet18, we use the "standard training setup" (the random seed is $111$) but with epoch to be $200$ to record more orderings. Here, we use two data loaders to the same training data – one is augmented for training, while the other is clean for ordering. In this way, we minimize the effect caused by other factors. For c-score.loss, we separate the training data into four parts and use $3/4$ of the data to train a new random initialized model – ResNet18, and record the value for the remainder $1/4$ data. A shorter epoch of $30$ is used for each training with cosine decaying learning rate (reaching zero after $30$ epoch).

For (anti-/random-)curriculum learning, we use the order c-score.loss right above. We follow the standard setup with step-type pacing. The hyper-parameters are $b = 0.2$ and $a = 0.8$. Algorithm 1 and 3 are used to measure the ordering during training.

**Figures in Section 4 and 5.** The experiments are explained partially in the main text. Due to the nature of the (anti-/random-) curriculum learning that has different iterations in an epoch (as we update the size of training data dynamically), we use the notation "iteration/step" instead of "epoch". We want to emphasize that when we decrease the number of training steps from $35200$ in Figure 5 to $17600$, $1760$ and $352$ in Figure 6, we implicitly accelerate the decrease in the learning rate as it eventually drops to zero within the steps we set. For a fair comparison, we set the configuration for each run the same as "standard training setup" (mentioned at the beginning of this section), including the data-augmentation, batch-size, optimizer (weight-decay and learning rate), and random seeds.

**Figure 10.** For FOOD101 and FOOD101N, we choose a batch size to be $256$ and use NVIDIA Tesla $8\times$V100 GPU for each experiment. The data augmentation is the same as the ImageNet training in PyTorch example (see Footnote 5). The optimizer is SGD with $0.9$ momentum, weight decay $1 \times 10^{-5}$, and a learning rate scheduler – cosine decay – with an initial value of $0.1$. We use half-precision for training.

## C  FIGURES

---

[6] https://github.com/geifmany/cifar-vgg

1 seed_111-arch_fc1000-adam
2 seed_222-arch_fc1000-adam
3 seed_111-arch_fc1000-adam.1
4 seed_333-arch_fc1000-adam
5 seed_333-arch_fc1000-adam.1
6 seed_333-arch_fc1000-scheduler_cosine
7 seed_111-arch_fc1000-scheduler_cosine
8 seed_111-arch_fc1000-scheduler_step2
9 seed_333-arch_fc1000-scheduler_step2
10 seed_333-arch_fc500-adam
11 seed_222-arch_fc500-adam
12 seed_111-arch_fc500-adam
13 seed_111-arch_fc500-adam.1
14 seed_222-arch_fc500-adam.1
15 seed_333-arch_fc500-adam.1
16 seed_333-arch_fc500-scheduler_cosine
17 seed_222-arch_fc500-scheduler_cosine
18 seed_111-arch_fc500-scheduler_cosine
19 seed_222-arch_fc500-scheduler_step2
20 seed_333-arch_fc500-scheduler_step2
21 seed_111-arch_fc500-scheduler_step2
22 seed_111-arch_vgg11-adam
23 seed_333-arch_vgg11-adam
24 seed_111-arch_vgg11-adam.1
25 seed_222-arch_vgg11-adam
26 seed_333-arch_vgg11-adam.1
27 seed_222-arch_vgg11-adam.1
28 seed_222-arch_vgg11-scheduler_cosine
29 seed_111-arch_vgg11-scheduler_cosine
30 seed_333-arch_vgg11-scheduler_cosine
31 seed_111-arch_vgg11-scheduler_step2
32 seed_222-arch_vgg11-scheduler_step2
33 seed_333-arch_vgg11-scheduler_step2
34 seed_333-arch_vgg19-adam
35 seed_111-arch_vgg19-adam
36 seed_222-arch_vgg19-adam
37 seed_333-arch_vgg19-scheduler_cosine
38 seed_111-arch_vgg19-scheduler_cosine
39 seed_222-arch_vgg19-scheduler_cosine
40 seed_333-arch_vgg19-scheduler_step2
41 seed_111-arch_vgg19-scheduler_step2
42 seed_222-arch_vgg19-scheduler_step2
43 seed_111-arch_vgg11_bn-adam
44 seed_333-arch_vgg11_bn-adam
45 seed_222-arch_vgg11_bn-adam
46 seed_333-arch_vgg11_bn-scheduler_cosine
47 seed_222-arch_vgg11_bn-scheduler_cosine
48 seed_111-arch_vgg11_bn-scheduler_cosine
49 seed_333-arch_vgg11_bn-scheduler_cosine.1
50 seed_222-arch_vgg11_bn-scheduler_cosine.1
51 seed_111-arch_vgg11_bn-scheduler_cosine.1
52 seed_333-arch_vgg11_bn-scheduler_step
53 seed_222-arch_vgg11_bn-scheduler_step
54 seed_111-arch_vgg11_bn-scheduler_step
55 seed_333-arch_vgg11_bn-scheduler_step.1
56 seed_111-arch_vgg11_bn-scheduler_step.1
57 seed_222-arch_vgg11_bn-scheduler_step.1
58 seed_333-arch_vgg19_bn-adam
59 seed_111-arch_vgg19_bn-adam
60 seed_222-arch_vgg19_bn-adam
61 seed_333-arch_vgg19_bn-scheduler_cosine
62 seed_111-arch_vgg19_bn-scheduler_cosine

63 seed_222-arch_vgg19_bn-scheduler_cosine
64 seed_222-arch_vgg19_bn-scheduler_cosine.1
65 seed_111-arch_vgg19_bn-scheduler_cosine.1
66 seed_333-arch_vgg19_bn-scheduler_step
77 seed_222-arch_vgg19_bn-scheduler_step
68 seed_111-arch_vgg19_bn-scheduler_step
69 seed_222-arch_vgg19_bn-scheduler_step.1
70 seed_111-arch_vgg19_bn-scheduler_step.1
71 seed_222-arch_densenet-adam
72 seed_333-arch_densenet-adam
73 seed_111-arch_densenet-adam
74 seed_111-arch_densenet-scheduler_cosine
75 seed_222-arch_densenet-scheduler_cosine
76 seed_333-arch_densenet-scheduler_cosine
77 seed_222-arch_densenet-scheduler_cosine.1
78 seed_111-arch_densenet-scheduler_cosine.1
79 seed_333-arch_densenet-scheduler_cosine.1
80 seed_111-arch_densenet-scheduler_step
81 seed_222-arch_densenet-scheduler_step
82 seed_333-arch_densenet-scheduler_step
83 seed_111-arch_densenet-scheduler_step.1
84 seed_333-arch_densenet-scheduler_step.1
85 seed_222-arch_densenet-scheduler_step.1
86 seed_222-arch_efficientnetB0-adam
87 seed_333-arch_efficientnetB0-adam
88 seed_111-arch_efficientnetB0-adam
89 seed_111-arch_efficientnetB0-scheduler_cosine
90 seed_333-arch_efficientnetB0-scheduler_cosine
91 seed_333-arch_efficientnetB0-scheduler_cosine.1
92 seed_111-arch_efficientnetB0-scheduler_cosine.1
93 seed_222-arch_efficientnetB0-scheduler_cosine
94 seed_222-arch_efficientnetB0-scheduler_cosine.1
95 seed_111-arch_efficientnetB0-scheduler_step
96 seed_111-arch_efficientnetB0-scheduler_step.1
97 seed_222-arch_efficientnetB0-scheduler_step
98 seed_333-arch_efficientnetB0-scheduler_step
99 seed_222-arch_efficientnetB0-scheduler_step.1
100 seed_333-arch_efficientnetB0-scheduler_step.1
101 seed_333-arch_preresnet18-adam
102 seed_111-arch_preresnet18-adam
103 seed_222-arch_preresnet18-adam
104 seed_222-arch_preresnet18-scheduler_cosine
105 seed_333-arch_preresnet18-scheduler_cosine
106 seed_333-arch_preresnet18-scheduler_step
107 seed_333-arch_preresnet50-adam

108 seed_111-arch_preresnet50-adam
109 seed_222-arch_preresnet50-adam
110 seed_111-arch_preresnet50-scheduler_cosine
111 seed_222-arch_preresnet50-scheduler_cosine
112 seed_333-arch_preresnet50-scheduler_cosine
113 seed_111-arch_preresnet50-scheduler_step
114 seed_333-arch_preresnet50-scheduler_step
115 seed_111-arch_preresnet50-scheduler_step.1
116 seed_222-arch_preresnet50-scheduler_step
117 seed_333-arch_wresnet28x10-adam
118 seed_111-arch_wresnet28x10-adam
119 seed_222-arch_wresnet28x10-adam
120 seed_222-arch_wresnet28x10-scheduler_cosine
121 seed_111-arch_wresnet28x10-scheduler_cosine
122 seed_111-arch_wresnet28x10-scheduler_cosine.1
123 seed_333-arch_wresnet28x10-scheduler_cosine
124 seed_222-arch_wresnet28x10-scheduler_cosine.1
125 seed_222-arch_wresnet28x10-scheduler_step
126 seed_111-arch_wresnet28x10-scheduler_step
127 seed_111-arch_wresnet28x10-scheduler_step.1
128 seed_222-arch_wresnet28x10-scheduler_step.1
129 seed_333-arch_wresnet28x10-scheduler_step
130 seed_222-arch_wresnet40x10-adam
131 seed_111-arch_wresnet40x10-adam
132 seed_333-arch_wresnet40x10-adam
133 seed_111-arch_wresnet40x10-scheduler_cosine
134 seed_222-arch_wresnet40x10-scheduler_cosine
135 seed_111-arch_wresnet40x10-scheduler_cosine.1
136 seed_333-arch_wresnet40x10-scheduler_cosine
137 seed_222-arch_wresnet40x10-scheduler_cosine.1
138 seed_333-arch_wresnet40x10-scheduler_step
139 seed_111-arch_wresnet40x10-scheduler_step
140 seed_111-arch_wresnet40x10-scheduler_step.1
141 seed_222-arch_wresnet40x10-scheduler_step
142 seed_222-arch_wresnet40x10-scheduler_step.1

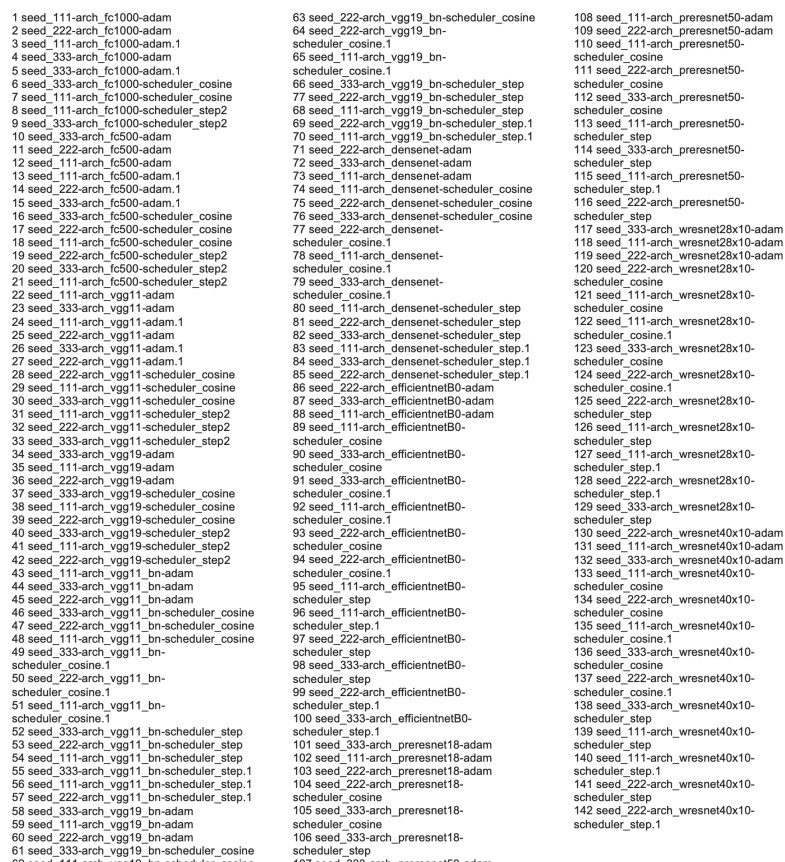

Figure 15: This figure describes the key hyper-parameters used for each of the columns in Figure 3. For example, the first top left: $1 seed\_111 - arch\_fc1000 - adam$ corresponds to the result of $1st$ column in Figure 3 where we use random seed 111 and fully-connected layers with Adam as an optimizer. The ambiguous one where there is $adam.1$, means that we use twice a longer time than those with configuration ends with $adam$.

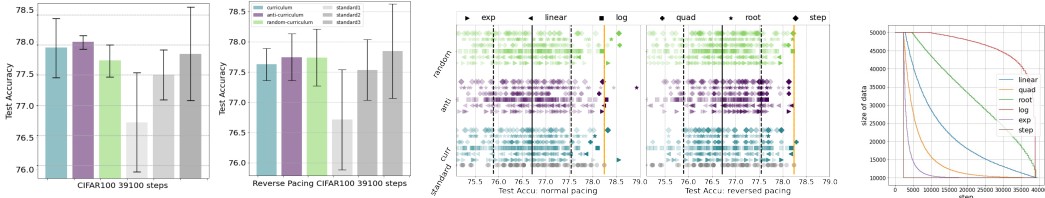

Figure 16: **Standard time training** for CIFAR100 with total 39100 steps. The training setup is different from the figures in the main sections, where we trained 45000 images and then use 5000 validation images to pick the best test accuracy. Here, we use the entire training samples 50000 and directly pick the best test accuracy from these 5000 test samples. Thus, the accuracy is higher in general. The left bar (dot) plot uses the normal increasing pacing functions. The right bar (dot) plot uses the reverse pacing functions (see the 5th plot for an example). That is, we start with a full data set and then slowly discard the images in $[g(t), N]$ as $g(t)$ is now decreasing. See Figure 5 for detailed description for each plot.

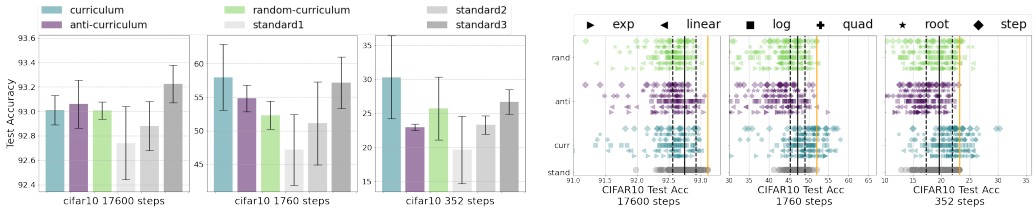

Figure 17: **Short time** training for CIFAR10 with total steps equal to 19550, 3910 and 391 (see x-label). See Figure 5 for detailed description for each plot.

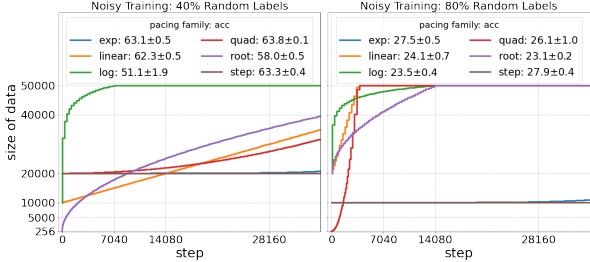

Figure 18: Top performing pacing functions for noisy training. Top performing pacing functions from the six families considered for CIFAR100 with 40% label noise (left) and 80% label noise (right).

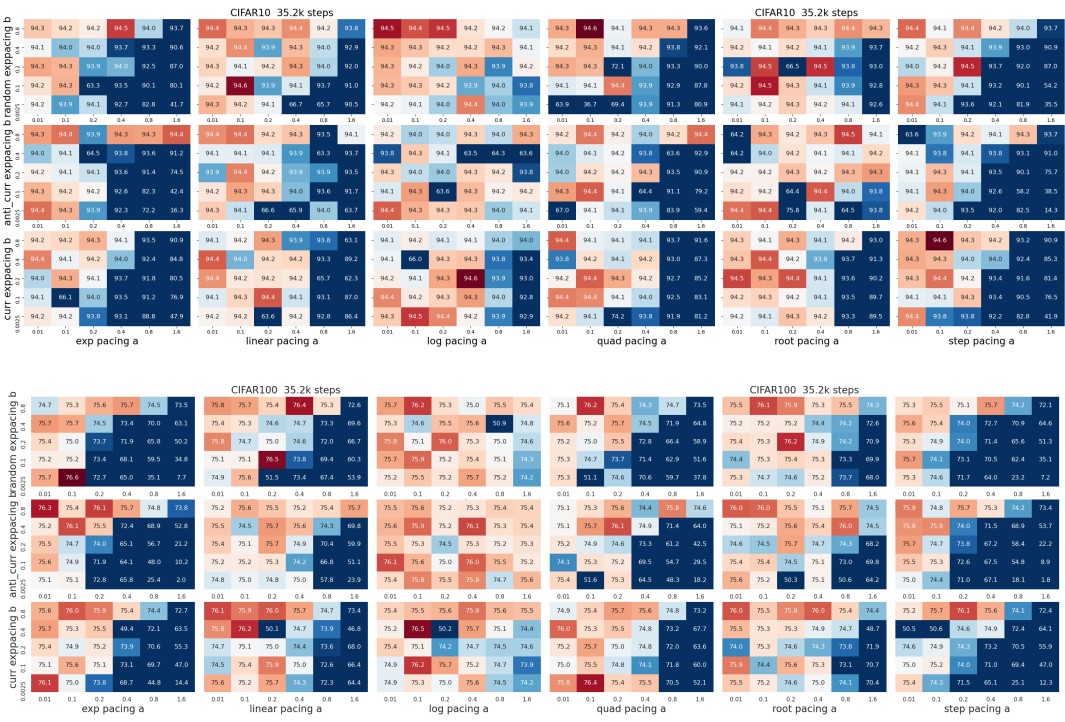

Figure 19: Standard time training with steps 35200. The top panel is for CIFAR10 and the bottom for CIFAR100. In each panel, we have $3 \times 6$ plots where each row from 1st to 3nd are random-curriculum, anti-curriculum, and curriculum learning (see the y-label). Each column means the type of pacing functions: from 1st to the 6th are exponential, linear, log quadratic, root, and step basic functions (see x label). Each plot inside a panel is a heat-map for the parameter $a \in \{0.01, 0.1, 0.2, 0.4, 1.0, 1.6\}$ at x-axis and the parameter $b \in \{0.0025, 0.1, 0.2, 0.4, 1.0\}$ at y-axis describing the best accuracy for the corresponding $(x, y) = (a, b)$. The dark blue to light blue means the lowest value to the median one, while the light red to dark red means the median to the highest one. So the darkest blue is the lowest, and the darkest red is the highest.

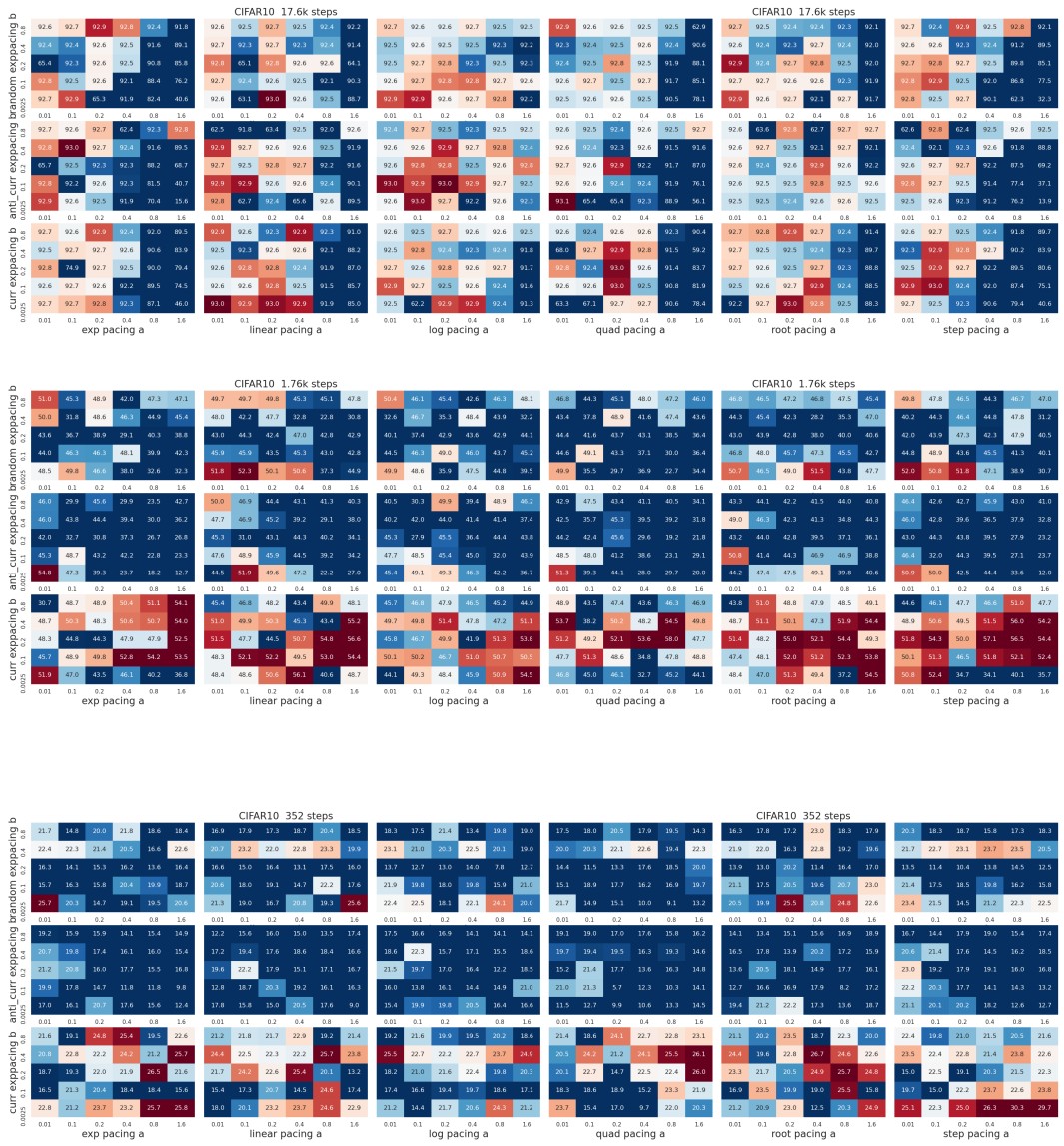

Figure 20: CIFAR10 short time training. Top, middle and bottom panels are total steps 17600, 1760 and 352.

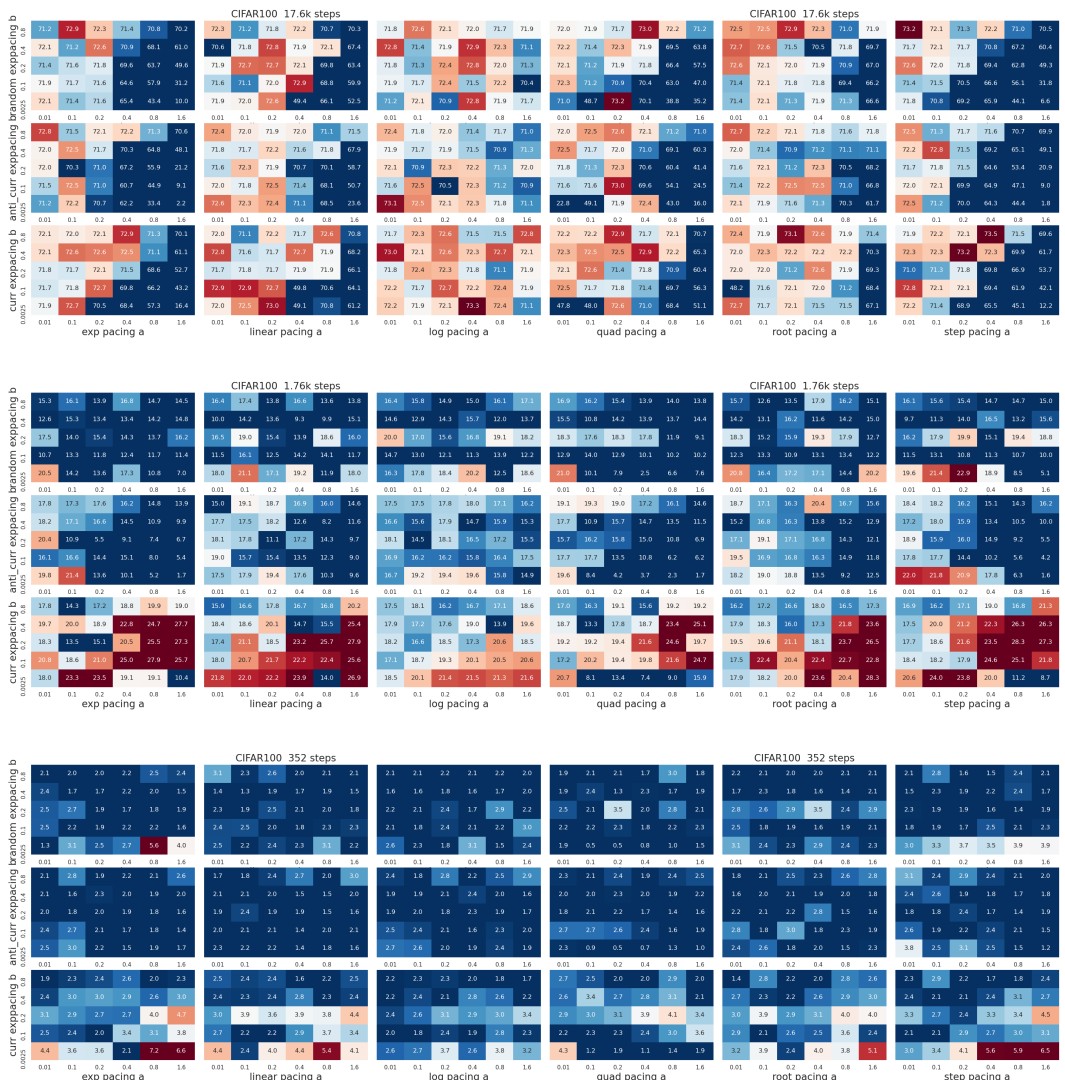

Figure 21: CIFAR100 short-time training. Top, middle and bottom panels are total steps 17600, 1760 and 352. See Figure 19 for detailed description.

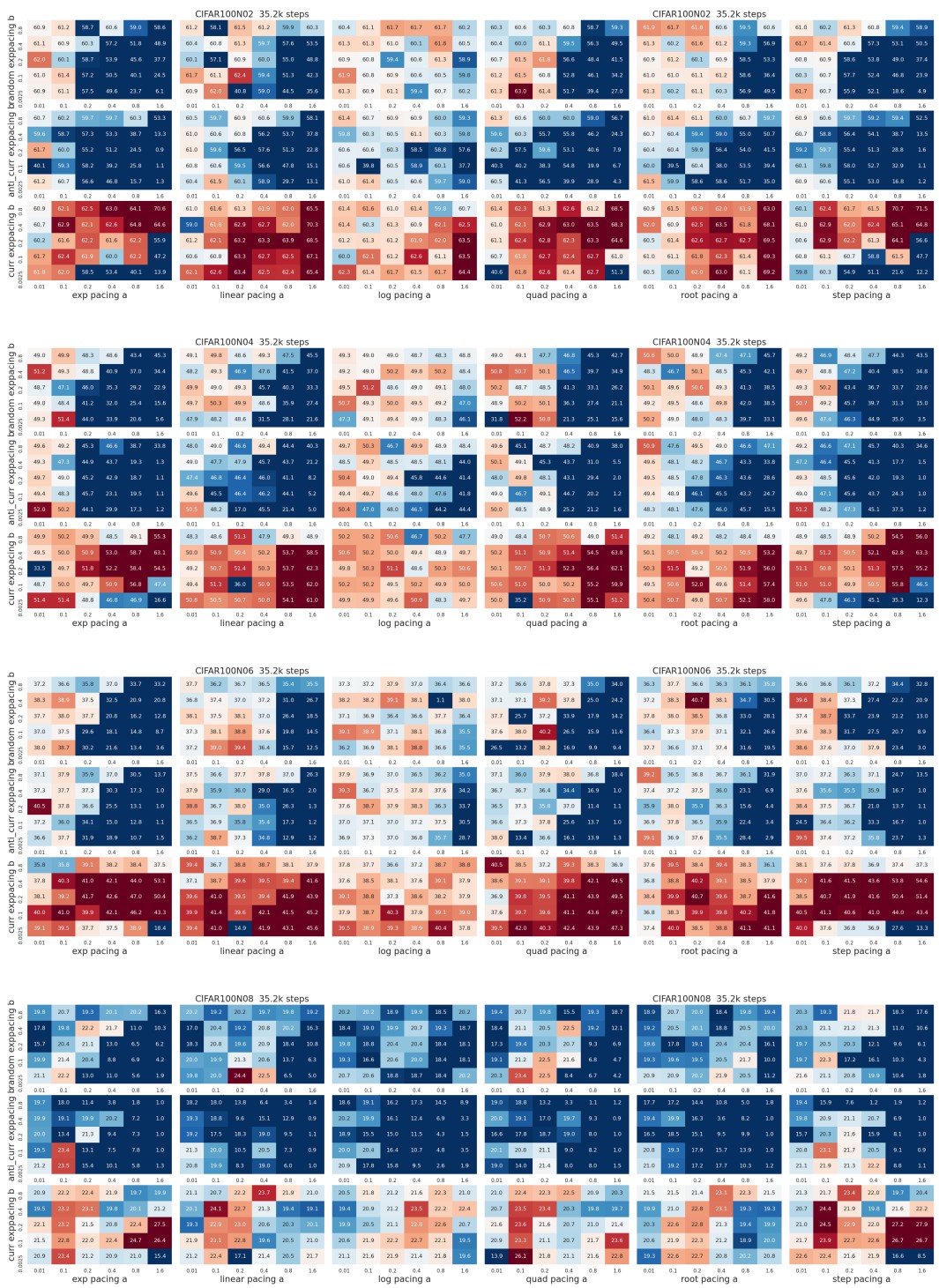

Figure 22: Noisy training for CIFAR100 with 35200 steps. From top to bottom plots are for 20%, 40%, 60% and 80% label noise; See Figure 19 for detailed description.

