# OpenReview forum: "When Do Curricula Work?"
_ICLR.cc/2021/Conference — ICLR 2021 Oral_

### Official Review · AnonReviewer4 · 2020-10-28
**A comprehensive empirical study with interesting results**

**Rating:** 7
**Confidence:** 3

**Review:**

This paper provides a comprehensive empirical study on the effects of the ordering of training samples in curriculum learning. The authors designed extensive experiments and obtained some interesting results: 1. the models with a similar architecture learn samples in a consistent order; 2. with enough iterations curriculum learning has no gain on performance comparing with random ordering; 3. curriculum learning outperforms others when the training time is limited; 4. curriculum learning is more robust with noisy samples.
In general, I think this is a quite practical work that could be beneficial to the community. The experiments are carefully designed and the results are sound, and the paper is well structured.

There are just some minor issues that may need some clarification:
1. As the authors mentioned in Sec.3, the random ordering is not as the same as i.i.d. training because it corresponds to dynamic training size. Isn't it more similar with bootstrapping? It would be interesting if the authors can have some discussion in this point of view.
2. The parameters of pacing function a, b should be introduced with the pacing function in the beginning of Sec.3 , as they were mentioned before Sec. 3.2 without explanation.
3. Subfigures in Fig.5 should have subcaptions.
4. In the experiments with noisy labels,  the best pacing functions ignore all noisy data, does it mean their gain of performance is simply from filtering out the noisy samples? Is there any other influence from the formulation of the pacing function itself?
5. In the paragraph under Fig.8, the last sentence 'a trend to start and maintain ...' is a bit confusing to understand, could authors clarify it a bit?

---

> ### Author Response · Authors · 2020-11-19
> **Response to Reviewer4**
>
>
> We thank the reviewer’s positive assessment and questions:
>
> 1.“the random ordering is not as the same as i.i.d. training because it corresponds to dynamic training size. Isn't it more similar with bootstrapping? some discussion in this point of view”
>
>       That is an interesting thought. Bootstrapping involves sampling the dataset with replacement. For a random curriculum, however, the order is fixed once at the beginning of training, and subsequently the dataset is drawn deterministically, given this initial random order, without replacement.
>
>
> 2."The parameters of pacing function a, b should be introduced with the pacing function in the beginning of Sec.3 , as they were mentioned before Sec. 3.2 without explanation."
>
>       Thank you for pointing this out. We have rearranged the discussion involving pacing functions so that all discussion of parameters is in Section 3.2 (see update version).
>
>
> 3." Subfigures in Fig.5 should have subcaptions."
>
>        Updated Fig 5.
>
>
>  4a). " In the experiments with noisy labels, the best pacing functions ignore all noisy data, does it mean their gain of performance is simply from filtering out the noisy samples?"
>
>      Yes, that is our observation.
>
> 4b). " Is there any other influence from the formulation of the pacing function itself?"
>
>      In the case of label noise we found that exponential and step pacing functions to be the best performing. In all cases studied (Figures 8 and 15) the exponential (blue) and step (grey) functions largely overlap and both appear to correspond to filtering out all noisy labels.
>
> 5."In the paragraph under Fig.8, the last sentence 'a trend to start and maintain ...' is a bit confusing to understand, could authors clarify it a bit?"
>
>      We apologize for the confusing wording. We meant that the pacing functions start with an initially small training dataset and that the dataset only grows gradually. We have rephrased the sentence as: “In the 2nd plot of Fig 8 (352 steps), we see many pacing functions start with a relatively small dataset size and maintain a small dataset for a large fraction of the total training time.”

---

### Official Review · AnonReviewer3 · 2020-10-28
**A nice paper**

**Rating:** 8
**Confidence:** 4

**Review:**

##########################################################################
Summary:

The paper provides a comprehensive analysis of the benefits of curriculum learning in different application scenarios. This includes investigating the phenomenon of implicit curricula, showing if the examples are learned in a consistent order across different architectures, and exploring the influences of explicit curricula in the standard and emulation settings. The paper empirically shows that curriculum learning has marginal benefits for standard training, but is helpful when the training time is limited or the training data is noisy.

##########################################################################

Reasons for score:

I vote for accepting the paper. I believe the analyses presented in the paper can be valuable for the community. I like the implicit curricula experiments, showing that the difficulty of an example is somewhat independent of the training method. Other empirical observations are also interesting. In general, I think the paper provides a satisfactory answer to the question raised in the paper title (when do curricula work?)

##########################################################################

Pros:

This paper has extremely comprehensive evaluations, examining the influence of curriculum learning (curriculum/anti-curriculum/random-curriculum) in diverse settings (standard/limited training time budget/noisy data). The methodology for the evaluations is technically sound;

The findings presented in the paper can be valuable for the community: (1) the difficulty of an example is somewhat independent of the training method; (2) curriculum learning provides little benefit for standard training, but help for limited time and noisy training;

The paper is well written. It is a thoroughly enjoyable experience to read the paper.

##########################################################################

Cons:

I found few weaknesses in the paper. I include a question below which I hope could be clarified:

The learned iteration of a sample is defined by the first epoch from which the prediction remains correct for all subsequent epochs. I wonder if there is any sample that is predicted correctly in earlier steps but incorrectly in later steps (e.g.  the forgettable examples). How to handle them in the implicit curricula experiment?



##########################################################################

Minor comments:

Page 16: two data loader → two data loaders

#########################################################################

Final recommendation:

I have read the authors' responses as well as the comments from my fellow reviewers. I would like to keep my rating of the paper (8).

---

> ### Author Response · Authors · 2020-11-19
> **Response to Reviewer3**
>
> We thank the reviewer for reading our paper and for the highly positive assessment.
>
> 1." I wonder if there is any sample that is predicted correctly in earlier steps but incorrectly in later steps…"
>
> Yes. This can indeed happen, and is one of the subjects of (Toneva et al. (2019) arXiv: 1812.05159). As you note, these examples are forgotten and thus the learned epoch for these is the same as an example which is not learned. To highlight these cases, we have now added Figure 10 to the appendix. There are samples predicted correctly during training but incorrectly at the end, though this behavior stabilizes as the learning rate is decayed towards the end of the training.
>
> 2." two data loader → two data loaders"
>
> Thank you for pointing out the typo. It is now corrected.

---

### Official Review · AnonReviewer2 · 2020-10-30
**Detailed, methodical, large-scale empirical evaluation of the impact of curriculum learning for image classification**

**Rating:** 8
**Confidence:** 3

**Review:**

Summary: The paper conducts a large-scale evaluation of the impact of curriculum learning (CL) in image classification. The paper progresses nicely through a sequence of well-thought research questions and experiments, with the key findings stated up front. In particular, the notion of "implicit curriculum" is shown to exist. Prior findings around when CL is helpful (limited training, label noise) are confirmed, which is nice. Overall, this methodical empirical evaluation comes away with a clear set of takeaways, empirically "summarizing" a lot of prior work on CL and the training of deep models. Some discussion about why CL helps when training is limited or data has label noise (or next steps) would strengthen the paper a bit more.

Strengths:
  + Extremely well-written and easy to read. Key findings are summarized and visualized up front.
  + Well-designed large-scale empirical investigation into important open questions for training image classifiers.

Areas for improvement
  - Can't think of too many. I suppose the paper could have included a bit more discussion into why the reduced training or label noise benefits from curriculum learning. I'm also curious to see how these findings compare with a similar study on sequence (especially text) data but as the paper mentions, it's outside the scope of this paper.
  - A pointer up front directing the reader to the appendix where "standard training" is defined would have been nice to have.

Questions:
  * On Page 5, I didn't follow the sentence "Given **these pacing functions**, we can now ask if the explicit curricula enforced by them can change the order in which examples are learned.". I agree that Fig 3-right shows that the difficulty ordering (e2d, rand, d2e) can change the order in which examples are learned when using the step pacing function. What other pacing functions are used in Fig 3-right? Is there a different interpretation of the sentence involving the pacing functions?

Minor comments
  - A few typos
    - Fig 2's caption ("ReseNet50", "EfficeientNet")
    - Page 6 ("CIAFR10")


UPDATE: I thank the authors for their detailed response and updated paper. I'm now more inclined to accept.

---

> ### Author Response · Authors · 2020-11-19
> **Response to Reviewer2**
>
> We thank the reviewer for taking the time to read our work and for the positive feedback and suggestions for improvement.
>
> 1." more discussion into why the reduced training or label noise benefits from curriculum learning..."
>
> Thanks for the feedback, as a step in this direction we have added a figure (Figure 9) and discussion. In summary:
>
> Figure 13 in the original submission plots the distribution of the loss-based c-score for CIFAR100 with and without label noise. For the distribution of clean CIFAR100, a significant number of images concentrate around zero.  However, the concentration slowly shifts to larger values as the label noise increases. In the updated version (Figure 9), we also computed c-score distributions for reduced time clean CIFAR10 and CIFAR100 training. We also find that the c-score distribution shifts to the right as training time is decreased. As c-score is intended to measure the difficulty of examples, this suggests that both label noise and reduced time training shift the c-score distributions towards more difficult examples and that curricula can help by focusing first on easier examples.
>
> 2."A pointer up front directing the reader to the appendix where "standard training" is defined would have been nice to have."
>
> We have now included this (see footnote 1 on page 2).
>
> 3."On Page 5, I didn't follow the sentence…"
>
> We thank you for pointing out the confusion. The paragraph "Given these pacing functions, ..."  has now been moved to Section 3.2 and the wording has been clarified (see updated version). We use only one pacing function - step with (a,b)=(0.8,0.2) in Figure 3-right.

---

### Comment · ~Xinshao_Wang1 · 2021-02-23
**Curricula learning versus  implicit example weighting in a loss function.**

Dear Xiaoxia Wu, Ethan Dyer, Behnam Neyshabur,

This is a great work to read. Congratulations!

BTW, we had two pieces of work which are highly related (In other words, curricula learning and example weighting are highly related). Our work studied the **implicit example weighting introduced by a loss's derivative and back-propagation** when optimising a model using gradient descent.
1. IMAE for Noise-Robust Learning: Mean Absolute Error Does Not Treat Examples Equally and Gradient Magnitude's Variance Matters https://arxiv.org/abs/1903.12141
2. Derivative Manipulation for General Example Weighting https://arxiv.org/abs/1905.11233

In a nutshell, a Curricula (i.e., learning order of data points), regardless of being implicit or explicit, will be influenced by the **implicit example weighting  introduced by a loss function**, especially when the derative is non-monotonic with respect to the loss value.


I think it will be very interesting to discuss them in the paper or study them together in the future work.
I am looking forward to your ideas, or further discussion if you are available.


Many thanks and kind regards.

---

### Decision · Program_Chairs · 2021-01-07
**Final Decision**

**Decision:**

Accept (Oral)

**Comment:**

This nice paper gives a better understanding of how Curriculum Learning (CL) affects image classification. In particular, it gives insight into cases such as noisy training data and limited training time. It shows that examples can be rated by difficulty to some extent, in that the order in which examples are learned seems to be consistent across runs. The paper is thorough and well-written.